# Effect of Composite Probiotics on Antioxidant Capacity, Gut Barrier Functions, and Fecal Microbiome of Weaned Piglets and Sows

**DOI:** 10.3390/ani14091359

**Published:** 2024-04-30

**Authors:** Jilang Tang, Mingchao Zhao, Wenyue Yang, Hong Chen, Yihan Dong, Qi He, Xue Miao, Jiantao Zhang

**Affiliations:** 1College of Veterinary Medicine, Northeast Agricultural University, Harbin 150030, China; kuyuren@126.com (J.T.); 17645474939@163.com (M.Z.); 13845494535@163.com (W.Y.); ch980815molihua@163.com (H.C.); 18624133998@163.com (Y.D.); 18846770949@163.com (Q.H.); 13204076720@163.com (X.M.); 2College of Veterinary Medicine, Hebei Agricultural University, Baoding 071001, China; 3Heilongjiang Key Laboratory for Laboratory Animals and Comparative Medicine, Harbin 150030, China

**Keywords:** probiotics, oxidative stress, piglets, sows, microbiota, correlation analysis

## Abstract

**Simple Summary:**

The stress response of pig herds is a major problem in the pig breeding industry, and exploring methods to solve pig herds stress is of great significance. Probiotics have good anti-stress effects. Therefore, this study intervened in weaned piglets and pregnant sows with a composite probiotic consisting of *lactobacillus plantarum*, *lactobacillus reuteri*, and *bifidobacterium longum*, and found that the composite probiotic can alleviate oxidative stress in weaned piglets and pregnant sows by improving the barrier function of intestinal tissues and altering the structure of fecal microbiota in the pig group.

**Abstract:**

This study investigated the efficacy of a composite probiotics composed of *lactobacillus plantarum*, *lactobacillus reuteri*, and *bifidobacterium longum* in alleviating oxidative stress in weaned piglets and pregnant sows. Evaluations of growth, oxidative stress, inflammation, intestinal barrier, and fecal microbiota were conducted. Results showed that the composite probiotic significantly promoted average daily gain in piglets (*p* < 0.05). It effectively attenuated inflammatory responses (*p* < 0.05) and oxidative stress (*p* < 0.05) while enhancing intestinal barrier function in piglets (*p* < 0.01). Fecal microbiota analysis revealed an increase in the abundance of beneficial bacteria such as *faecalibacterium*, *parabacteroides*, *clostridium*, *blautia*, and *phascolarctobacterium* in piglet feces and *lactobacillus*, *parabacteroides*, *fibrobacter*, and *phascolarctobacterium* in sow feces, with a decrease in harmful bacteria such as *bacteroides* and *desulfovibrio* in sow feces upon probiotic supplementation. Correlation analysis indicated significant negative associations of *blautia* with inflammation and oxidative stress in piglet feces, while *treponema* and *coprococcus* showed significant positive associations. In sow feces, *lactobacillus*, *prevotella*, *treponema*, and *CF231* exhibited significant negative associations, while *turicibacter* showed a significant positive association. Therefore, the composite probiotic alleviated oxidative stress in weaned piglets and pregnant sows by modulating fecal microbiota composition.

## 1. Introduction

The stress response of pig herds is currently a major concern in the swine farming industry, posing a significant threat to pig health and welfare and causing substantial economic losses. Modern intensive farming practices, early weaning techniques for piglets, and tethered housing for sows have been implemented to improve production efficiency in pig farms but have also induced significant stress reactions in pig herds. The premature separation of piglets from sows and changes in diet and social environment, coupled with the incomplete development of various organ systems in piglets, make it difficult for piglets to adapt to complex environments, resulting in reduced feed intake, damage to the intestinal barrier, and dysregulation of gut microbiota and the immune system [1]. These factors ultimately result in post-weaning diarrhea and growth restrictions in piglets. Tethered housing of sows alters their behavior, physiology, and immune response [2]. In modern pig farms, continuous genetic selection for high-yielding and lean offspring leads to increased catabolism in sow organisms. Enhanced catabolism promotes the production of reactive oxygen species (ROS), leading to increased oxidative stress. In the late stages of sow pregnancy, they experience severe catabolism, with plasma concentrations of α-tocopherol and retinoid decreasing by 56% and 57%, respectively, compared to day 110 of pregnancy and day 30 of pregnancy [3]. This heightened catabolic state and oxidative stress result in DNA damage within the sow’s body [4]. The damage caused by oxidative stress, combined with heat stress resulting from high temperature conditions, may be a key factor in sow abortion [5]. Therefore, improving the stress response in pig herds is crucial for swine production.

Probiotics as a means to counteract stress-induced damage in pig herds is also a promising option. The beneficial effects of probiotics on stress response have been confirmed. Probiotics can synergistically promote the expression of glucocorticoid receptors with N-3 polyunsaturated fatty acids (PUFA), reduce adrenocorticotropic hormone expression, and lower corticosterone levels in the blood [6]. *Lactobacillus acidophilus* can decrease cortisol levels in the blood of suckling piglets challenged with lipopolysaccharides, and reduce the levels of white blood cells, neutrophils, and lymphocytes in the blood [7]. Furthermore, *bacillus* with high antioxidant capacity can regulate the Nrf2/Keap1 signaling pathway and reduce ROS generation to counteract oxidative damage induced by H_2_O_2_ in porcine intestinal epithelial cells (IPEC) [8]. These research findings confirm the potential of probiotics in combating oxidative stress in pigs.

Studies have shown that dietary supplementation of probiotics can improve the stress status of sows, reduce oxidative stress and inflammatory reactions, and improve the nutritional metabolism of piglets by altering the gut microbiota [9]. The gut microbiota has an impact on the host’s stress state. Stress signals originating from the gut can modulate pig behavior and affect their susceptibility to pathogens. Therefore, utilizing the host-microbe interaction to address the current pressures of stress and disease in pigs has emerged as a novel and effective approach [10]. Formula milk fermented with lactic acid bacteria can increase the abundance of *lachnospira* and *anaerorhabdus furcosa* genera in the intestine of piglets, which are positively correlated with the production of short-chain fatty acids (SCFAs) and contribute to improving the stress response during weaning in piglets [11]. *Lactobacillus reuteri* is capable of reducing the abundance of *clostridium_sensu_stricto_1*, *terrisporobacter*, *blautia* and *streptococcus* in the feces of growing pigs, while increasing the abundance of *christensenellaceae_R-7_group*, *rikenellaceae_RC9_gut_group*, *bifidobacteriaceaeg_bifidobacterium*, and *lactobacillus*. Additionally, it enhances the concentration of SCFAs in the feces, elevates the levels of immunoglobulin G, immunoglobulin M, interferon-gamma, and interleukins 2, 4, and 10 in serum, and reduces the concentration of pro-inflammatory cytokines in serum, thereby improving the immune performance of growing pigs [12]. *Bifidobacterium longum*, in combination with human milk oligosaccharides (HMO), can reduce the abundance of *clostridium sensu stricto 1* and *clostridium perfringens* in the intestines of pre–term piglets, thereby ameliorating necrotizing enterocolitis (NEC) in pre–term piglets [13]. These studies collectively suggest the promising application of probiotics in alleviating stress and disease pressure in pig populations.

Probiotics have been widely adopted as the preferred choice for mitigating stress in pig farming due to their ability to regulate the gut microbiota, broad availability, affordability, and safety. However, the precise mechanisms by which probiotics exert their effects are not yet fully understood. In this study, a composite probiotic consisting of three commercially available probiotic strains with antioxidative stress properties was chosen to investigate the mechanism of probiotics in combating stress in pig herds, taking into account the changes in fecal microbiota. The findings aim to provide guidance for the application of probiotics in pig production.

## 2. Materials and Methods

### 2.1. Bacterial Strains

The strains of *lactiplantibacillus plantarum* BNCC185392, *lactobacillus reuteri* BNCC192190, and *bifidobacterium longum* BNCC185354 were purchased from Beina Co., Ltd. (Beijing China). The fermented products of different strains were mixed and stored in a refrigerator at 4 °C for future use. The population of each probiotic was not less than 1 × 10^9^ CFU/g.

### 2.2. Experimental Animals and Sample Collection

This study was conducted at the sanhua pig farm of northeast agricultural university. In this study, 80 healthy weaned piglets (breeds: yorkshire × landrace × duroc or landrace × yorkshire × duroc, both genders; average weaning age: 24.3 ± 1.33 days) were selected and grouped in pens of eight with ten pens, and randomly divided into two groups, i.e., the control group (C1 group) and the probiotic-treated group (T1 group), with five pens per group, totaling forty piglets per group. The fermented probiotic solution, prepared by mixing and stirring at a ratio of 1:1:1, was added to the feed. Each liter of the fermented probiotic solution was mixed with 1000 kg of feed to create a diluted mixture (the population of each probiotic was not less than 1 × 10^9^ CFU/kg). The piglets were fed with this mixture for 7 days before and after weaning, continuously for a period of 14 days. In both groups, 16 piglets were selected and weighed before and after the feeding period. At the end of the feeding period, 10 randomly selected piglets from each group underwent collection of blood samples via jugular venipuncture and collection of feces. Then, these 10 piglets were euthanized, and colon tissue samples were collected for subsequent experiments. After blood collection, serum was separated using a low-temperature centrifuge (Eppendorf, Hamburg, Germany, 5000 r/min, centrifuged for 15 min), then aliquoted into 100 mL EP tubes and stored at −20 °C. After isoflurane anesthesia, euthanasia was performed by exsanguination from the axillary artery, the colon was ligated, and approximately 1 g of feces was aseptically collected and transferred into 1 mL cryovials. The fecal samples were then placed in liquid nitrogen for preservation. Then, collected duodenal tissues were stored at −80 °C for subsequent sectioning.

Additionally, 100 healthy late-term pregnant sows (breeds: yorkshire × landrace or landrace × yorkshire, Sow parity 2–5, 75–80 days of gestation) were selected and randomly divided into two groups: the control group (C2 group) and the probiotic-treated group (T2 group). The C2 group was fed a conventional diet, while the T2 group received feed supplemented with the probiotic fermented solution, which was mixed and stirred at a ratio of 1:1:1. The dosage of the fermented solution was 1 L per 1000 kg of feed (the population of each probiotic was not less than 1 × 10^9^ CFU/kg). The sows were fed with this mixture continuously for 30 days until they were transferred to farrowing crates. After the feeding period, 10 randomly selected sows from each group underwent collection of blood samples via jugular venipuncture and collection of feces. The method for collecting serum from sows was identical to that of piglets. During feces collection, strict aseptic techniques were employed, and only freshly expelled feces were collected. Approximately 1 g of feces per sow was promptly transferred into 1 ml cryovials. Fecal samples were initially preserved using dry ice on site and then stored in liquid nitrogen.

The weaned piglets were housed in environmentally controlled, fully slatted floor pens with adjustable temperature and humidity, maintained at 22–25 °C and 60–70% humidity. Piglets were raised in farrowing crates before weaning and transferred to piglet nursery pens after weaning. They were provided with ad libitum access to feed troughs and water nipples. Pregnant sows were housed in environmentally controlled, partially slatted floor pens with free access to water, and were fed manually twice daily with a total of 3 kg of complete gestation diet per day. Pre-weaned piglets were fed a complete pre–starter diet before weaning, followed by a complete starter diet post-weaning. The nutritional composition of the piglet diet included crude protein 16%, crude ash 7%, crude fiber 4%, lysine 1.3%, calcium 1.2%, total phosphorus 5%, and moisture 14%. Pregnant sows were fed a complete gestation diet containing crude protein 15%, crude fiber 10%, crude ash 10%, calcium 1%, phosphorus 0.5%, lysine 0.5%, and moisture 14%. All complete diets were purchased from Da Bei Nong Co., Ltd. (Beijing China). The experiment was approved by the Animal Welfare Committee of Northeast Agricultural University (NEAUEC20220121). All procedures were conducted in accordance with animal welfare guidelines.

### 2.3. Serological Analysis

The stored serum was retrieved from the −20 °C freezer and allowed to thaw at room temperature for 15 min. Then, use an ELISA assay kit to detect the levels of Interleukin–1ß (IL–1ß), Interleukin–6 (IL–6), Tumor necrosis factor–α (TNF–α), malondialdehyde (MDA), glutathione (GSH), superoxide dismutase (SOD), Adrenocorticotropic hormone (ACTH), Cortisol (COR), and Epinephrine (EPI) in the serum, while adhering to the instructions provided by the ELISA assay kit (Jiangsu Jingmei Biocompany, Yancheng, China).

### 2.4. Histological Analysis

The duodenal samples of piglets were treated with a 4% paraformaldehyde solution to fix them and then embedded in paraffin. Next, the samples were sectioned into 4 μm slices using a microtome (Zhejiang Jinhua Kedi Instrument Equipment, Jinhua, China) and stained with hematoxylin and eosin. Finally, the samples were viewed under light microscopy (BX–FM; Olympus Corp, Tokyo, Japan) at a magnification of 200×.

### 2.5. Immunohistochemistry

The paraffin sections of the duodenum were treated through a series of steps to prepare them for staining. Initially, they were dewaxed using xylene and then rehydrated using a decreasing concentration of ethanol. To permeabilize the sections, 0.5% Triton X–100 was used for 15 min. Afterward, three washes with HBSS were applied to remove any remaining reagents. Next, 3% H_2_O_2_ was used to eliminate catalase. To conduct ZO–1 (Zonula occludens-1) staining, the sections were subjected to overnight incubation at 4 °C using anti–ZO–1 rabbit pAb (1:200, ABclonal, Wuhan, China) in a humidified box. Conversely, for HO–1 (Heme oxygenase–1) and p-p65 (Phosphorylated Recombinant Transcription Factor p65) staining, anti–HO–1 and p-p65 rabbit pAb (1:100, Wanlei, Shenyang, China) was used to incubate the sections overnight at 4 °C. Finally, the antibody-stained sections were treated with goat anti-rabbit antibody (1:200, Boster, Wuhan, China) for one hour at 37 °C and washed. Slice the sample and observe it under a light microscope (BX–FM; Olympus Corp, Tokyo, Japan), then collect data using imageJ for processing.

### 2.6. mRNA Detection in the Duodenum

The duodenal samples underwent RNA extraction using the RNA Simple Total RNA Kit from Tiangen Biotech Co., Ltd. in Beijing, China. In this study, 2 µg of total RNA was used to synthesize complementary DNA (cDNA) via reverse transcription, following the manufacturer’s instructions for the TransScript^®^ All–in–One First–Strand cDNA Synthesis SuperMix for PCR from Transgen Biotech Co., Ltd. The resulting cDNA was then used in amplification (40 cycles) with forward and reverse primers, included β–actin (beta cytoskeletal actin), Claudin1, Muc2 (Mucin 2), TNF–α, IL–1β, IL–6, NF–κB (Nuclear Factor-kappa B) (Table 1), SYBR Green SuperReal PreMix Color, and ddH_2_O. The fluorescence intensity was monitored throughout the PCR process using the LightCycler^®^480 from Roche in Shanghai, China, and cycle threshold (Ct) values were obtained.

### 2.7. 16S rRNA Microbiota Analysis

Total genomic DNA was extracted from each fecal sample using the OMEGA Soil DNA Kit (M5635–02) (Omega Bio–Tek, Norcross, GA, USA). The V3–V4 region of the bacterial 16S rRNA gene was amplified using the forward primer 338F (5′–ACTCCTACGGGAGGCAGCA–3′) and reverse primer 806R (5′–GGACTACHVGGGTWTCTAAT–3′). We used Illumina NovaSeq for sequencing.

The study employed microbiome bioinformatics using QIIME2 2019.4, with slight adjustments from the official tutorial. Initially, raw sequence data were demultiplexed utilizing the demux plugin, followed by primer cutting through the cutadapt plugin. Afterwards, the sequences underwent quality filtering, denoising and merging, and chimera removal using the DADA2 plugin. Non-singleton amplicon sequence variants (ASVs) were then aligned with mafft. Subsequently, ASV-level alpha diversity indices such as Chao1, Faith’s PD, Good’s coverage, Shannon, Simpson, and Observed species were calculated based on the ASV table in QIIME2 and displayed as box plots. Beta diversity analysis was carried out to explore microbial communities’ structural variation across samples leveraging Bray–Curtis metrics and visualized via principal coordinate analysis (PCoA). MEGAN and GraPhlAn were utilized to visualize the taxonomic composition and abundance. The default parameters were leveraged for linear discriminant analysis effect size (LEfSe) to identify differentially abundant taxa across groups. Additionally, UPGMA clustered the samples according to the species composition data’s Euclidean distance to create a species composition heat map.

### 2.8. Statistical Analysis

GraphPad Prism 9.5 (GraphPad Software, Inc., San Diego, CA, USA) was used to analyze the data. Group comparisons were performed using unpaired *t* test. If the data are normally distributed but have heterogeneous variances, Welch’s correction *t*-test was employed. For non-normally distributed data, the Mann–Whitney U test, a non-parametric test, was utilized. The correlation coefficients between floras and oxidative stress factors and inflammatory factors were analyzed using spearman statistical method. Moreover, a *p* < 0.05 was considered statistically significant.

## 3. Result

### 3.1. The Effects of a Composite Probiotic on Piglet Growth Performance 

The results showed that (Table 2): there were no significant differences in piglet body weight between groups C1 and T1 before and after the experiment (*p* > 0.05). However, the average daily gain in group T1 was significantly higher than that in group C1 (*p* < 0.05).

### 3.2. The Effects of Composite Probiotics on Inflammatory Response in Weaned Piglets and Pregnant Sows

The serum test results showed that the IL–1ß, IL–6, and TNF–α level of group C2 pregnant sows was significantly higher than that of group T2, with an extremely significant difference (*p* < 0.0001, *p* < 0.05, and *p* < 0.01) (Figure 1A–C). The IL–1, IL–1ß, and TNF–α levels in the serum of group C1 were significantly higher than those in group T1 (*p* < 0.0001, *p* < 0.001, and *p* < 0.0001) (Figure 1D–F). The mRNA detection results showed that the relative expression levels of IL–1ß, IL–6, TNF–α, and NF–κB mRNA in group T1 were significantly reduced compared with those in group C1 (*p* < 0.05, *p* < 0.01, *p* < 0.0001, and *p* < 0.0001) (Figure 1G–J).

### 3.3. The Effects of Composite Probiotics on Oxidative Stress in Weaned Piglets and Pregnant Sows

We conducted measurements of stress indicators in the serum of weaned piglets and pregnant sows, and the results showed that the levels of ACTH and COR in the serum of weaned piglets fed with composite probiotics were significantly lower compared to the control group (*p* < 0.0001) (Figure 2A,B). Similarly, the levels of ACTH and COR in the pregnant sows fed with composite probiotics were significantly reduced compared to the control group (*p* < 0.0001) (Figure 3A,B). The EPI level in group T1 was significantly lower than in group C1 (*p* < 0.01) (Figure 2C), while there were no significant differences in EPI levels between group T2 and C2 (*p* > 0.05) (Figure 3C). Composite probiotics significantly reduced the MDA levels in the serum of weaned piglets and pregnant sows (*p* < 0.0001) (Figure 2A and Figure 3A), and increased the levels of SOD and GSH (*p* < 0.05 and *p* < 0.0001) (Figure 2B,C and Figure 3B,C).

### 3.4. The Results of Duodenal Barrier in Weaned Piglets

After performing HE staining on piglet duodenal tissues (Figure 4I), we found that the arrangement of villous epithelial cells in weaned piglets was relatively loose, and some epithelial cells were shed. In contrast, the villous epithelial cells in piglets fed with composite probiotics were arranged more tightly, and the villous structure was relatively complete. Immunohistochemical results showed that the mean optical density of ZO–1, a tight junction protein, was significantly higher in group T1 compared to group C1 (*p* < 0.01) (Figure 4II,V); the mean optical density of HO–1, an antioxidant protein, was also significantly higher in group T1 compared to group C1 (*p* < 0.01) (Figure 4III,VI); and the mean optical density of p–p65, a protein associated with inflammation, was significantly lower in group T1 compared to group C1 (*p* < 0.01) (Figure 4IV,VII). mRNA detection results of MUC2 and Claudin 1, proteins associated with mucus and tight junctions, respectively, showed that the mRNA expression of both genes in the duodenal tissues of piglets in group T1 were significantly higher than those in group C1 (*p* < 0.001) (Figure 4VIII,IX).

### 3.5. Fecal Microbiota Analysis

Subsequently, we investigated the fecal microbiota of weaned piglets and pregnant sows. The Chao1, Faith-pd, Good-coverage, Shannon, Simpson, and Observed-species indices exhibited no significant changes (*p* > 0.05) in the weaned piglet test group (T1) compared with the control group (C1), and there was no difference in α–diversity analysis between the two groups (Figure 5A). In contrast, the Chao1, Faith–pd, and Observed–species indices in the treatment group (T2) were significantly higher than those of the control group (C2) (*p* < 0.001), while the Shannon index was significantly higher in T2 compared to C2 (*p* < 0.001). The Good–coverage index was significantly lower in T2 than C2 (*p* < 0.001), and there were no significant differences between the Simpson indices in T2 and C2 (*p* > 0.05) (Figure 6A). The β–diversity principal coordinate analysis results showed that the C1 and T1 samples were widely separated, and that the microbial community structures were significantly different between the two groups (Figure 5B). The distances between the C2 and T2 groups were also distinguishable, and their microbial compositions were different (Figure 6B). The species composition results at the genus level showed that the abundance of *prevotella*, *lactobacillus*, *roseburia*, *oscillospira*, *treponema*, *coprococcus*, and *lachnospira* was higher in the C1 group than in the T1 group. Conversely, *phascolarctobacterium*, *blautia*, *collinsella*, *ruminococcaceae_ruminococcus*, *parabacteroides*, *SMB53*, *CF231*, *clostridiaceae_clostridium*, and *gemmiger* showed lower abundance in the C1 group and higher abundance in the T1 group (Figure 5C). *Lactobacillus*, *prevotella*, *treponema*, *ruminococcaceae_ruminococcus*, *YRC22*, *parabacteroides*, *fibrobacter*, *phascolarctobacterium*, and *CF231* displayed higher abundance in the T2 group than in the C2 group, while *oscillospira*, *turicibacter*, *bacteroides*, *caloramator*, *paludibacter*, *desulfovibrio*, and *coprococcus* exhibited lower abundance in the T2 group compared to C2 (Figure 6C). The LEfSe analysis results showed that (Figure 5D): the annotated differential phyla in group T1 were firmicutes and actinobacteria, while in group C1, the annotated differential phyla were spirochaetes, proteobacteria, TM7, deferribacteres, and cyanobacteria. At the class level, the annotated differential taxa in group T1 were erysipelotrichi and coriobacteriia, while in group C1, they were spirochaetes, betaproteobacteria, TM7–3, verrusco–5, epsilonproteobacteria, deferribacteres, and mucispirillum. At the order level, the annotated differential taxa in group T1 were erysipelotrichales, coriobacteriales, i025, and rhodospirillales, while in group C1, they were spirochaetales, tremblayales, CW040, GMD14H09, pasteurellales, elusimicrobiales, campylobacterales, sphaerochaetales, mycoplasmatales, and deferribacterales. At the family level, the annotated differential taxa in group T1 were erysipelotrichaceae, S24–7, chitinophagaceae, and peptostreptococcaceae, while in group C1, they were prevotellaceae, spirochaetaceae, F16, RF16, pasteurellaceae, RFP12, elusimicrobiaceae, heliconacteraceae, sphaerochaetaceae, mycoplasmataceae, and deferribacteraceae. At the genus level, the annotated differential taxa in group T1 were *gruhacterium*, *blautia*, *collinsella*, *veillonella*, *subdoligranulum*, *ruminococcus*, *gemniger*, *dorea*, *bulleidia*, *butyricimonas*, *adlercreutzia*, *lactococcus*, *coprobacillus*, and *sharpea*, while in group C1, they were *prevotella*, *coprococcus*, *streptococcus*, *lvsobacter*, *haemophilus*, *serratia*, *helicobacter*, *sphaerochaeta*, *mycoplasma*, and *acidaminococcus*. The clustered heatmap of the top 20 differential taxa at the genus level showed an increased abundance of beneficial genera, such as *faecalibacterium*, *parabacteroides*, *clostridium*, *blautia*, and *phascolarctobacterium* in group T1 (Figure 5E). LEfSe analysis results revealed that (Figure 6D) in the feces of pregnant sows, the differentially abundant phyla in the C2 group were firmicutes, lentisphaerae, and synergistetes, while in the T2 group, they were bacteroidetes, cyanobacteria, elusimicrobia, fibrobacteres, and spirochaetes. At the class level, the differentially abundant classes in the C2 group were acidobacteria_6, clostridia, betaproteobacteria, synergistia, RF3, and opitutae, while in the T2 group, they were bacteroidia, 4C0d_2, elusimicrobia, fibrobacteria, erysipelotrichi, alphaproteobacteria, and spirochaetes. At the order level, the differentially abundant orders in the C2 group were turicibacterales, clostridiales, burkholderiales, desulfovibrionales, syntrophobacterales, synergistales, and ML615J_28, whereas in the T2 group, they were bacteroidales, elusimicrobiales, fibrobacterales, lactobacillales, erysipelotrichales, RF32, GMD14H09, sphaerochaetales, spirochaetales, and anaeroplasmatales. Furthermore, the differentially abundant families in the C2 group were rikenellaceae, carnobacteriaceae, turicibacteraceae, christensenellaceae, dehalobacteriaceae, eubacteriaceae, desulfovibrionaceae, syntrophobacteraceae, and synergistaceae, while in the T2 group, they were corynebacteriaceae, chitinophagaceae, prevotellaceae, RF16, S24_7, elusimicrobiaceae, fibrobacteraceae, lactobacillaceae, streptococcaceae, veillonellaceae, erysipelotrichaceae, alcaligenaceae, sphaerochaetaceae, spirochaetaceae, and anaeroplasmataceae. At the genus level, the differentially abundant genera in the C2 group were *adlercreutzia*, *butyricimonas*, *paludibacter*, *alistipes*, *turicibacter*, *mogibacterium*, *sedimentibacter*, *clostridium_celatum*, *dehalobacterium*, *anaerofustis*, *clostridium*, *allobaculum*, *comamonas*, *oxalobacter*, *serratia*, and *synergistes*, whereas in the T2 group, they were *corynebacterium*, *CF231*, *prevotella*, *fibrobacter*, *lactobacillus*, *streptococcus*, *helcococcus*, *sarcina*, *anaerostipes*, *butyrivibrio*, *lachnospira*, *roseburia*, *faecalibacterium*, *anaerovibrio*, *phascolarctobacterium*, *bulleidia*, *coprobacillus*, *sutterella*, *janthinobacterium*, *moraxella*, *sphaerochaeta*, *treponema*, and *anaeroplasma*. Clustering heatmap analysis of the top 20 differentially abundant genera (Figure 6E) revealed that the abundance of beneficial bacteria such as *lactobacillus*, *parabacteroides*, *fibrobacter*, and *phascolarctobacterium* increased in the T2 group, while harmful bacteria such as *bacteroides* and *desulfovibrio* decreased.

### 3.6. Correlation Analysis

In order to further investigate the influence of fecal microbiota on oxidative stress in weaned piglets’ pregnant sows, we performed association analysis between the differential microbiota aa? t the genus level of the top 20 abundant genera and serum indicators. The results of fecal microbiota association analysis in weaned piglets showed that *blautia* was significantly negatively correlated with IL–1ß, IL–6, TNF–α, MDA, ACTH, COR, and significantly positively correlated with GSH. *Treponema* was significantly positively correlated with IL–6, TNF–α, MDA, ACTH, and EPI. *Coprococcus* was significantly positively correlated with IL–1ß, TNF–α, MDA, and ACTH and significantly negatively correlated with SOD and GSH. *CF231* was significantly negatively correlated with IL-6 and COR and significantly positively correlated with GSH (Figure 7A). The results of fecal microbiota association analysis in pregnant sows showed that *lactobacillus* was significantly negatively correlated with IL–1ß, TNF–α, MDA, ACTH, COR, and significantly positively correlated with GSH. *Prevotella* was significantly negatively correlated with IL–1ß, IL–6, TNF–α, MDA, ACTH, COR, and significantly positively correlated with GSH. *Turicibacter* was significantly positively correlated with IL–1ß, IL–6, TNF–α, MDA, ACTH, COR, and significantly negatively correlated with GSH. *CF231* was significantly negatively correlated with IL–1ß, TNF–α, MDA, ACTH, COR, and significantly positively correlated with GSH (Figure 7B).

## 4. Discussion

Based on the significant impact of oxidative stress on weaned piglets and pregnant sows, this study utilized these two pig categories to evaluate the effects of probiotics in alleviating oxidative stress. Probiotics have been demonstrated to enhance feed conversion efficiency and ameliorate oxidative stress status in pig herds. Supplementing the feed with *lactobacillus plantarum* ZLP001 can improve feed conversion rate in piglets, while reducing the levels of MDA in the serum and increasing the concentrations of SOD, GSH, and catalase [14]. Supplementation of dietary with *lactobacillus plantarum* and *saccharomyces cerevisiae* during pregnancy and lactation in sows can alleviate oxidative stress and inflammatory responses, and improve piglet’s nutritional metabolism by altering the gut microbiota, thereby promoting growth and addressing health issues in piglets [9]. Our results confirmed that the composite probiotics can promote feed conversion by alleviating oxidative stress in piglet herds.

The composite probiotics not only exert antioxidative stress effects but also demonstrate significant intervention effects on inflammation. Inflammation is often interrelated with oxidative stress [15]. Moreover, the anti-inflammatory properties of probiotics have been well-documented in numerous studies. These anti-inflammatory properties may be attributed to the exopolysaccharides (EPS) produced by probiotics. EPS derived from *lactobacillus* have been shown to effectively reduce the levels of pro-inflammatory cytokines IL–1ß, IL–6, and TNF–α in colitis mice [16]. The anti-inflammatory properties of *lactobacillus*, in turn, lead to alterations in the intestinal microbiota [17]. Furthermore, the polysaccharides derived from *bifidobacterium longum* not only promote the growth of macrophages and enhance their phagocytic ability but also regulate the NF–κB signaling pathway to reduce the expression of pro-inflammatory cytokines [18]. The phenomena observed in our study are consistent with these findings.

The anti-inflammatory and antioxidative properties of composite probiotics also influence intestinal barrier function. Oxidative stress factors and pro-inflammatory cytokines can damage intestinal epithelial cells. Reactive oxygen species (ROS) generated by oxidative stress can induce damage-associated molecular patterns (DAMPs), stimulating the NF–κB signaling pathway to produce inflammatory factors, thereby damaging mitochondria-induced cell apoptosis pathways [19]. Pro–inflammatory cytokines can promote the redistribution of tight junction proteins such as occludin and claudin–1 in intestinal epithelial cells to enhance intercellular permeability, and induce apoptosis of intestinal epithelial cells, thus compromising intestinal barrier function [20]. These factors may contribute to intestinal barrier damage in oxidative stress pig herds. The anti-inflammatory and antioxidative properties of composite probiotics are targeted towards protecting barrier function. Our study also confirmed, at the genetic and protein expression levels of intestinal barrier proteins, that composite probiotics could maintain intestinal barrier function through anti-inflammatory and antioxidative mechanisms.

The fecal microbiota analysis of weaned piglets revealed no significant differences in α–diversity. This outcome may be attributed to the physiological conditions of piglets. In modern industrial farming, piglets are often weaned at an early age, typically at 3–4 weeks or even earlier. During this period, the nervous system of piglets is not fully developed [21], and thus, they are less sensitive to environmental changes compared to adult pigs. Additionally, our experimental period was relatively short, which may have limited the effectiveness of the composite probiotic in intervening with the piglet gut microbiota. The changes in the fecal microbiota structure of pigs due to probiotic supplementation were evident. The β–diversity of piglets indicated differences in the microbiota structure between the C1 and T1 groups. At the taxonomic level, feeding probiotics to weaned piglets resulted in a decrease in the abundance of *bacteroides*, *prevotella*, and *treponema*, and an increase in the abundance of *blautia*, *ruminococcus*, *clostridium*, and *lachnospira*, which were also the differentiating taxa between these two groups. Previous studies have shown significant changes in the gut microbiota structure of piglets during the weaning period. Weaning leads to a decrease in the abundance of Firmicutes, an increase in the abundance of Bacteroidetes, and an increase in the abundance of *prevotella* and *lactobacillus*. Furthermore, metagenomic sequencing revealed a higher abundance of functional gene clusters associated with oxidative stress response in the bacterial metagenome of weaned piglets compared to suckling piglets. Maternal supplementation of probiotics can reduce the levels of MDA in the serum of nursing Bamahu pigs, while also altering the composition of the colonic microbiota in piglets [9]. Due to the complexity of the gut microbiota, the results of this experiment also reflect that probiotics improve oxidative stress in weaned piglets by altering the gut microbiota, although the composition of the microbiota differs. *Bacteroides*, commonly found in the human gut, is an opportunistic pathogen causing extraintestinal infections. This bacterium exhibits a strong response to oxidative stress, thrives in oxygen-containing tissues such as the peritoneal cavity, and can lead to the formation of abscesses [22]. *Prevotella* is a common symbiotic bacterium in the gut, and some bacteria within this genus can transform into pathogens when immune system balance is disrupted, leading to host inflammatory responses [23]. Infection with *treponema* is associated with swine ear necrosis syndrome [24]. Moreover, *treponema* species are also involved in porcine skin ulcers [25]. In this experiment, the abundance of these potentially pathogenic bacteria significantly changed under the influence of probiotics. *Blautia* is significantly negatively correlated with host obesity, cancer, and various inflammatory diseases. It can produce bacteriocins, exhibit antimicrobial effects, regulate glucose and lipid homeostasis to reduce obesity-related inflammatory complications, and exert anti-inflammatory effects by producing SCFAs in the intestine. Therefore, *blautia* is a potential probiotic [26]. *Ruminococcus*, *clostridium*, and *lachnospira* all produce SCFAs and have certain probiotic effects. The SCFAs they produce can be directly utilized by intestinal tissues to generate energy and have anti-inflammatory and antioxidant effects [27]. In this experiment, it was found that the composite probiotic enhanced the tolerance of weaned piglets to oxidative stress by reducing the abundance of harmful bacteria and increasing the abundance of beneficial bacteria in feces.

In this experiment, the impact of probiotics on pregnant sows was found to be more significant than that on piglets. From the perspectives of α and β diversity, there were evident differences in fecal microbiota composition among groups of pregnant sows. The results of this study indicated a decrease in harmful bacteria such as *bacteroides* and *desulfovibrio* in the T2 group. *Desulfovibrio* is an opportunistic pathogen associated with ulcerative colitis and chronic periodontitis [28]. *Phascolarctobacterium* has a high colonization rate in the human gut and is capable of metabolizing succinate into propionate. Acetate is also a by–product of its metabolism, which decreases with aging [29]. Despite having certain antibiotic resistance, *parabacteroides* can metabolize carbohydrates to produce short-chain fatty acids (SCFAs), making it a candidate probiotic for the next generation [30]. *Fibrobacter*, a major bacterium involved in fiber digestion in the gastrointestinal tract of herbivores, is a potent fiber-digesting bacterium capable of degrading lignocellulosic fibers [31]. The increase in abundance of beneficial bacteria such as *lactobacillus*, *parabacteroides*, fibrobacter, and *phascolarctobacterium* in the T2 group may constitute a major factor in the efficacy of probiotics observed in this experiment. Due to the higher content of fiber in the feed of pregnant sows, we speculate that the alleviation of oxidative stress in pregnant sows may be associated with the increased abundance of cellulose-metabolizing bacteria in feces [32]. The use of probiotics in pig production to improve performance, blood–related parameters, and the resilience of pig herds has been proven to be an effective approach [33].

The correlation analysis between serum indicators and fecal microbiota reveals a relationship between the alterations in fecal microbiota and oxidative stress and inflammation markers in the bloodstream. However, the mechanisms by which gut microbiota influences stress responses remain uncertain. The interplay between the hypothalamic–pituitary–adrenal axis (HPA) and gut microbiota influences the host’s stress status. Under stress conditions, serum corticosterone levels were shown to be 2.8 times higher in germ-free (GF) rats compared to specific pathogen-free (SPF) rats [34]. Moreover, there was a negative correlation between serum corticosterone concentration and the abundance of fecal microbiota *akkermansia* and rikenella [35]. Transplanting fecal samples from depressed patients into GF rats resulted in increased depressive and anxious behavior in the rats due to the transmissible effects of gut microbiota [36]. Studies on the compositional changes in gut microbiota during stress have yielded highly variable results, with most findings derived from rodent models. Nevertheless, there is a commonality in the decreased abundance of beneficial bacteria such as *lactobacillus* and *bifidobacterium* [35]. The decrease in abundance of these beneficial bacteria significantly impacts the host’s inflammation under stress conditions [37]. It has been demonstrated that probiotics exert an indirect influence on the host’s stress status through immunomodulation. For example, the probiotic strain *lactobacillus rhamnosus* JB–1 can alleviate anxiety-like behavior by attenuating stress-induced dendritic cell activation [38]. In conclusion, our results also suggest that a combination of probiotics can alleviate inflammation and oxidative stress in stressed pig herds by modulating the fecal microbiota. However, the underlying mechanisms involved require further investigation.

## 5. Conclusions

Composite probiotic can enhance the intestinal barrier function of weaned piglets and alleviate oxidative stress and inflammatory responses in weaned piglets and pregnant sows. Moreover, there are correlations between fecal microbial composition and inflammation cytokine and oxidative stress markers.

## Figures and Tables

**Figure 1 animals-14-01359-f001:**
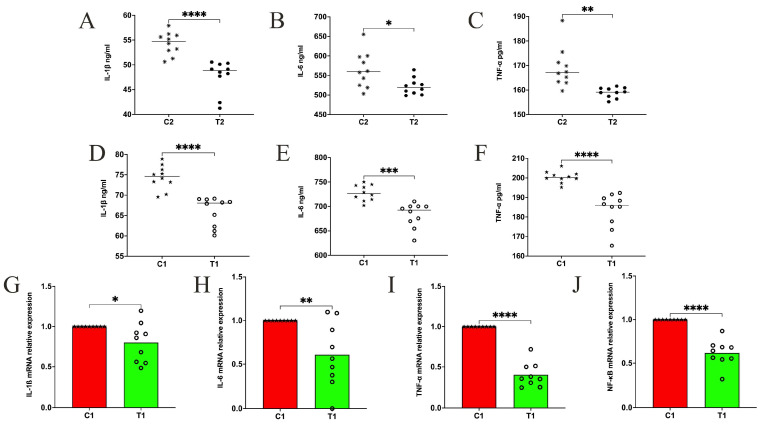
Serum inflammatory cytokines of sows and piglets and the relative mRNA expression of inflammatory factors in duodenal tissue of piglets. (**A**–**C**) Inflammatory cytokines Interleukin-1ß (IL–1ß), Interleukin-1 (IL–6), and Tumor necrosis factor–α (TNF–α) in sows’ serum. N = 10. (**D**–**F**) Inflammatory cytokines IL–1ß, IL–6 and TNF–α in piglets’ serum. N = 10. (**G**–**J**) The relative mRNA expression levels of IL–1ß, IL–6, TNF–α and Nuclear Factor–kappa B (NF–κB) in the piglets’ duodenum. N = 9. * *p* < 0.05, ** *p* < 0.01, *** *p* < 0.001 and **** *p* < 0.0001.

**Figure 2 animals-14-01359-f002:**
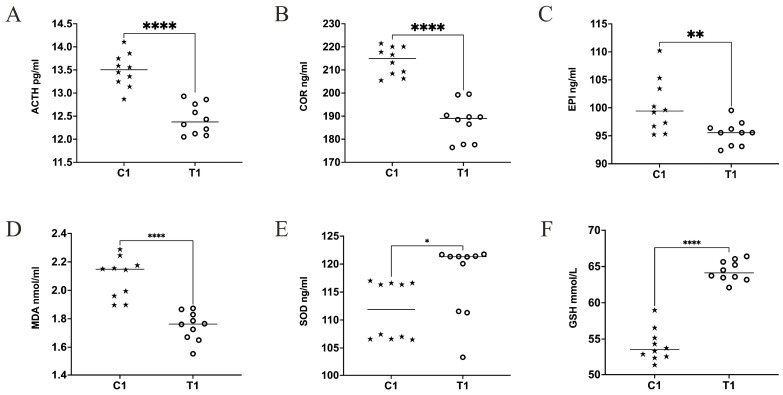
Serum factors of stress in piglets. (**A**) Adrenocorticotropic hormone (ACTH). (**B**) Cortisol (COR). (**C**) Epinephrine (EPI). (**D**) Malondialdehyde (MDA). (**E**) Superoxide dismutase (SOD). (**F**) Glutathione (GSH). n = 10. * *p* < 0.05, ** *p* < 0.01, and **** *p* < 0.0001.

**Figure 3 animals-14-01359-f003:**
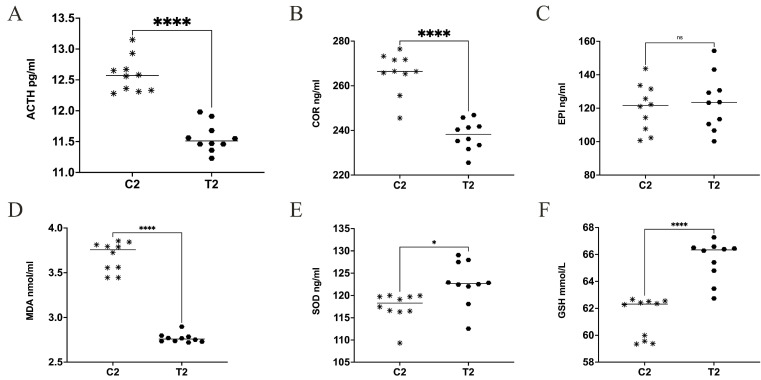
Serum factors of stress in sows. (**A**) Adrenocorticotropic hormone (ACTH). (**B**) Cortisol (COR). (**C**) Epinephrine (EPI). (**D**) Malondialdehyde (MDA). (**E**) Superoxide dismutase (SOD). (**F**) Glutathione (GSH). n = 10. ^ns^ *p* > 0.05, * *p* < 0.05, and **** *p* < 0.0001.

**Figure 4 animals-14-01359-f004:**
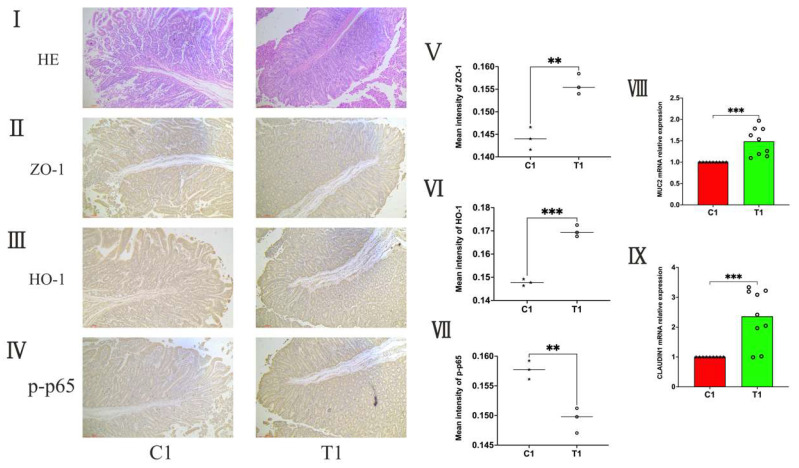
Duodenal barrier of piglets. (**I**) Duodenal histopathology was examined by HE staining, representative photomicrographs are presented at 200× magnification. n = 3. (**II**–**IV**) Immunohistochemistry staining of ZO–1, HO–1 and p–p65, representative photomicrographs are presented at 200× magnification. n = 3. (**V**–**VII**) fluorescence density calculation of Zonula occludens–1 (ZO–1), Heme oxygenase–1 (HO–1), and Phosphorylated Recombinant Transcription Factor p65 (p–p65). n = 3. (**VIII**,**IX**) The relative mRNA expression levels of mucin Mucin 2 (MUC2) and tight junction protein Claudin 1 in the piglet’s duodenal tissue. n = 9. ** *p* < 0.01 and *** *p* < 0.001.

**Figure 5 animals-14-01359-f005:**
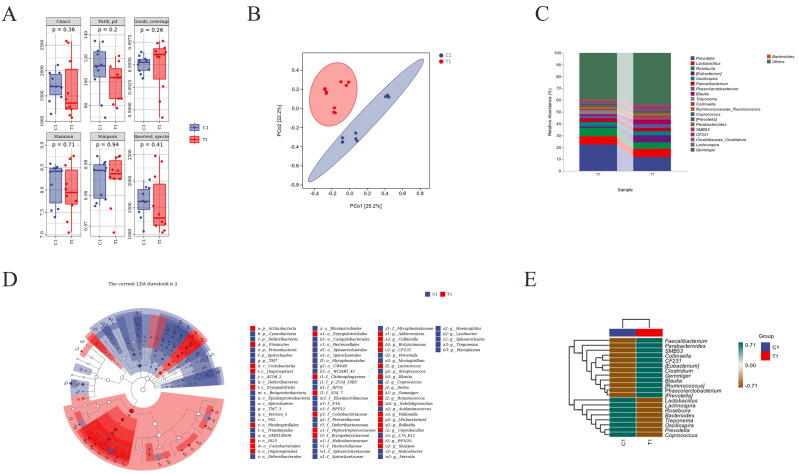
Analysis of piglets’ fecal microbiota. (**A**) α−diversity analysis. (**B**) Principal co−ordinates analysis (PCoA). (**C**) Community bar plots at the genus level. (**D**) Taxonomic cladogram of linear discriminant analysis effect size (LefSe) analysis. (**E**) Cluster heatmap analysis of the top 20 differentially abundant genera. n = 10.

**Figure 6 animals-14-01359-f006:**
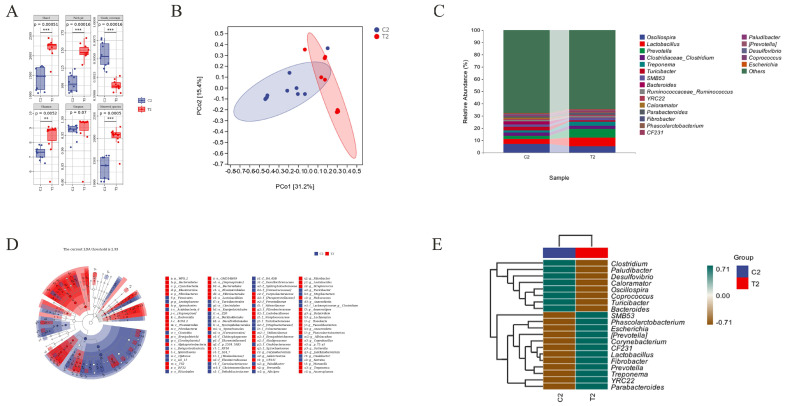
Analysis of sows’ fecal microbiota. (**A**) α−diversity analysis. (**B**) Principal co−ordinates analysis (PCoA). (**C**) Community bar plots at the genus level. (**D**) Taxonomic cladogram of linear discriminant analysis effect size (LefSe) analysis. (**E**) Cluster heatmap analysis of the top 20 differentially abundant genera. n = 10.

**Figure 7 animals-14-01359-f007:**
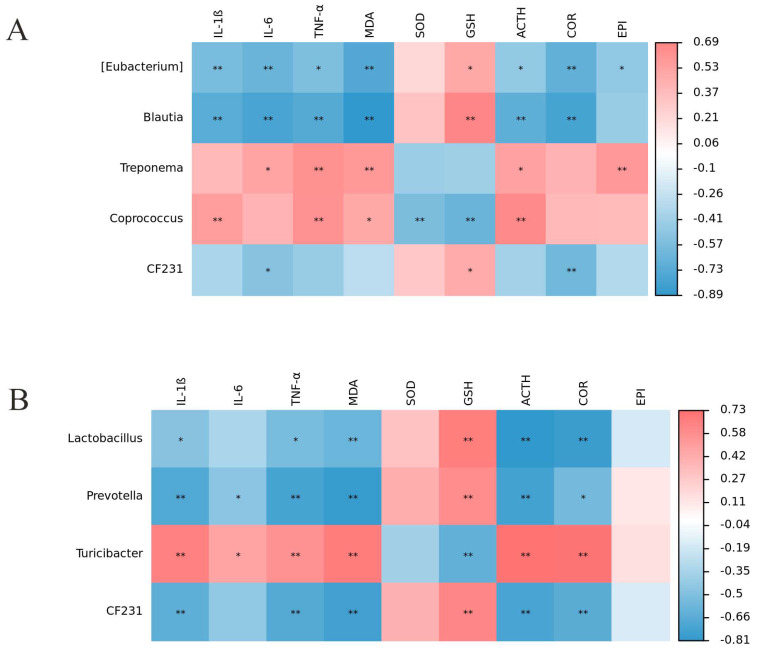
Correlation analysis between fecal microbiota and serum cytokines. (**A**) Correlation heatmap between microbiota and serum cytokines in piglets. (**B**) Correlation heatmap between microbiota and serum cytokines in sows. Red indicates positive correlation, blue indicates negative correlation, darker colors represent stronger correlation, and * indicates significance. n = 10. * *p* < 0.05 and ** *p* < 0.01.

**Table 1 animals-14-01359-t001:** Primer sequence.

Target Gene	Forward	Reverse
beta cytoskeletal actin (β–actin)	TCTGGCACCACACCTTCT	TGATCTGGGTCATCTTCTCAC
Claudin1	AGATTTACTCCTACGCTGGT	GCACCTCATCATTCCAT
Mucin 2 (Muc2)	CTGCTCCGGGTCCTGTGGGA	CCCGCTGGCTGGTGCGATAC
Tumor necrosis factor–α (TNF–α)	CCAGCTCTTCTGCCTACTGC	GCTGTCCCTCGGCTTTGAC
Interleukin-1ß (IL–1β)	AGTGAGAAGCCGATGAAGA	CATTGCACGTTTCAAGGATG
Interleukin-6 (IL–6)	CCTCTCCGGACAAAACTGAA	TCTGCCAGTACCTCCTTGCT
Nuclear Factor-kappa B (NF–κB)	GCGGGGACTACGACCTGAAT	GCACGGTTGTCAAAGATGGG

**Table 2 animals-14-01359-t002:** Piglets’ body weight.

	C1	T1
Start body weight (kg)	7.12 ± 0.40	7.13 ± 0.49
End body weight (kg)	8.42 ± 0.47	8.67 ± 0.35
Average daily gain (kg)	0.13 ± 0.02	0.15 ± 0.02 ^a^

Note: ^a^ indicates a significant difference compared to C1 group (*p* < 0.05).

## Data Availability

Data will be made available on request. Sequences data in this study were uploaded in NCBI SRA database. The SRA accession number is PRJNA1060644.g.

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
