# Peer review of "Effect of Composite Probiotics on Antioxidant Capacity, Gut Barrier Functions, and Fecal Microbiome of Weaned Piglets and Sows"

_animals, 2024, doi:10.3390/ani14091359_

Round 1
Reviewer 1 Report
Comments and Suggestions for Authors
This study found that composite probiotics can alleviate oxidative stress in weaned piglets and pregnant sows by improving the composition of fecal microbiota. This is a very valuable study and can be published in “Animals”. However, I have some comments:
1. The abstract is not well organized; the contents are incomplete. It would be feasible to emphasize the importance of oxidative stress in weaned piglets and pregnant sows.
2. Introduction: The content of the last two paragraphs should be carefully organized. For example, it would be better to move this content “Therefore, probiotics as a means to counteract stress-induced damage in pig herds is also a promising option” to the next paragraph.
3. The figures are not aesthetically pleasing. In particular, the size of label (ABCD) is not suitable for the size of figure.
4. In Figure4, the IHC results were not obvious, and no ruler was added.
5. In order to evaluate each gene’s prognostic value in predictive signature, survival curves were developed. To evaluate whether composite probiotics alleviate oxidative stress in weaned piglets by improving fecal microbiota composition, fecal microbiota analysis was used. Actually, we did not find any significant differences in theα-diversity analysis (Fig 5A). How to explain this phenomenon? Please add a limitation in the discussion section.
6. The author's language in the section of “Discussion” should be as concise as possible. And a limitation should be added in the section of “Discussion”
7. The section of “conclusion” is missing from the paper.
8. What breed of weaned piglets and pregnant sows were used in the experiment? Do the results of this study have the same results in other breeds of pigs?
9. Are the 80 healthy weaned piglets used in the experiment of the same sex? Does the study need to consider the effect of sex on the experiment?
10. The C1 group in Figure 1 G-J and Figure 4 VIII-IX where the quantitative results are calculated is 1, please confirm how the quantitative PCR results are calculated.
Comments on the Quality of English LanguageSome sentences in the manuscript are obscure and difficult to understand. There are some grammar errors. English expression needs to be further improved.
Author Response
Dear reviewer,
Our manuscript was revised according to your comments, and the itemized response to comments is attached. We marked all the changes in red in the revised manuscript. Many thanks for your suggestion. I am so sorry to bring you so much trouble because of our careless. Thanks very much for your attention to our paper. Once again, thank you for your help to our paper processing.
Best wishes!

Reviewer 2 Report
Comments and Suggestions for Authors
This research presents results on the study of a mixture of probiotic bacteria and its effect on reducing oxidative stress in piglets and pregnant sows. The findings of this research could provide guidance on the application of probiotics in swine production. However, the manuscript presents some considerations that must be addressed before publication.
Line 87: Change sentence to past tense.
Materials and methods.
Section 2.2.
Was there no approval of the experimental protocol by an ethics committee? Indicate the housing conditions for piglets and pregnant sows. Were they housed in controlled temperature and humidity conditions? What type of facilities, type of floor, feeder, waterer? Was fed provided ad libitum? etc.
Section 2.4 (lines 118-122).
In the sample collection section (2.2), it was never mentioned that the piglets were slaughtered and an intestinal samples were collected. Where? How?. What portion of the intestine was the tissue sample collected? Were the animals slaughtered to obtain this sample or was it a biopsy? It is only was mentioned that blood and feces samples were collected.
Line 120: With what equipment the intestinal tissue sample was sectioned? With a cryostat?
Statistical Section (2.8) The normality of the data was not verified?
The T test for two independent samples assumes normality and homoscedasticity of the data.
Also indicate in this section what type of correlation was made between the microbiome and the oxidative stress variables. Was it a Pearson correlation?.
Results Section
Table 2. Average daily gain instead Averagi daily weight gain.
Verify the significance in the daily gain of the piglets. The means differ by 20 grams (0.13 vs 0.15) and the standard error is 30 grams. This cannot be significant. Check this out. Is it standard deviation or standard error?
Title of figure 1. ....cytokines in the intestinal tissue of piglets and sows. (Include Sows).
Titles of figures 2 and 3: Are cortisol, ACTH, Epinephrine, MDA, and SOD cytosines? Adjust the titles of figures 2 and 3.
Reviewer 3 Report
Comments and Suggestions for Authors
The authors present a study on the supplementation of piglets and pregnant sows with a mix of probiotics. They investigated the effects of the supplementation on serum levels of some cytokines, the intestinal barrier of the piglets, and fecal microbiota. The introduction provides sufficient information to justify the experimental approach. However, the wording should be re-considered in some instances (see below). The presentation of the results is clear and supported by figures and tables. Again, some points should be addressed. The discussion is rather lengthy and starts with a section on effects of yeasts etc. which was not investigated here. Accordingly, there is some potential to shorten the discussion. Following points should be addressed in a revision:
(I) The experimental setup must be explained: According to the 3R rules, the number of experimental animals must be restricted to a number necessary to obtain meaningful results. In this study 80 piglets and 100 sows were treated and just 9-10 animals per group were selected for subsequent analyses. What justifies this approach? More importantly, there is no slaughter described of any animal (2.2). However, sections 2.4-2.6 describe histological and mRNA analyses in piglets’ intestines. How were the samples collected? Where is the mandatory ethical approval when slaughter of piglets is involved?
(II) Section 2.6 describes the mRNA analysis. How were the results calculated? There is just ß-actin included in the primer table apparently serving as reference gene. The use of a single reference gene does not conform to the MIQE guidelines for qPCR. Please give more details. Moreover, the presentation of the qPCR results is unusual giving the control group with 9 points set on the value “1”.
(III) The authors use frequently the term “stress of pig herds”. What should this mean? Here only some stress-related parameters for a small group of animals was investigated. Therefore, the introduction and the abstract should be re-worded in some parts.
(IV) L126/127: The antibodies must be explained. What does ZO-1, HO-1 and p-p53 mean? What specific changes should be addressed with the selected antibodies? Moreover, the photographs presented in figure 4 (II – IV) do not reflect the data in the graphs V-VII. The diffuse shadows visible in some pictures point to an unspecific staining of some areas rather than to specific marking of (sub)cellular structures of the intestine.
(V) L371: I doubt that increased backfat thickness is a sign of better feed concversion. On the contrary, efficient pigs convert feed energy into muscle or the growth of the fetuses rather than in backfat.
(VI) The discussion should be focused on the own results with relevant literature. Remove sections dealing with effects of yeast etc.
Comments on the Quality of English LanguageEnglish language is good - only minor issues were detected.
Round 2
Reviewer 3 Report
Comments and Suggestions for Authors
All points were addressed in a reasonable manner.
Author Response
Dear reviewer:
Our manuscript was revised according to your comments, and the itemized response to comments is attached. We marked all the changes in red in the revised manuscript. Thank you very much for your efforts in reviewing our manuscript progress. Please refer to the following content for our responses.
Please remove the keyword 'attenuate'.
Answer: We have removed the keyword “attenuate”. See line 33.
Please add piglets and sows to the keywords.
Anwser:We have added piglets and sows to the keywords. See line 33.
L81 SCFA was already defined on L76.
Answer: We have made corrections by removing the full name of SCFAs. See line 82.
L103 the products were stored in which condition?
Answer:We have provided information related to products storage. See lines 103-104.
L108 Please add how weaned piglets were assigned to treatments and how they were housed (randomized complete block design or completely randomized design), housed in 3 - 4 pigs per pen?
This is important for statistical analysis for model.
Answer:Dear reviewer, we adopted a completely randomized design. We have made corresponding modifications in the manuscript regarding the grouping, penning arrangements, and the number of piglets per group.See lines 109-111.
Please correct the title of 2.7 section.
Answer:Based on your suggestions, we have corrected the errors in the title of 2.7. See line 198.
Regarding the abbreviation in the figures, please follow author's guidelines.
Answer: According to the author guidelines, we have added the full names of abbreviations in the titles of the figures and tables. See table 1 and lines 252-256, 270-271, 275-276,296-297,371-372,and 375-376.
Regarding the backfat thickness of sows, unless there are data for initial backfat thickness, the data at the end of the study can't be interpreted correctly since if the sows already had higher backfat thickness when they were allotted (at initial), they would have higher backfat thickness at the end. So please include initial backfat thickness and calculate the backfat thickness change. If there is a significant difference in the change, then it could result from probiotics that could increase the nutrient reserve as the backfat thickness is within an acceptable range for late gestating sows.
Answer:Dear reviewer, your insights are indeed accurate. We acknowledge your concerns. Due to an oversight in our experimental design, we failed to measure the backfat thickness of the sows at the initiation of the experiment. Consequently, we are unable to provide the initial backfat datas of the sows and hence cannot present the datas on the changes in sows’ backfat thickness. After thorough discussion among all authors, we propose to present basic nutritional data post-experimentation as representative of sow conditions. However, if you find this inappropriate, we are open to your suggestions. We are willing to remove sow backfat-related datas from the manuscript and eliminate any references to backfat throughout the manuscript. We await your guidance on whether the removal of sow backfat data is necessary.
please check italic for microbes in L413-423.
Answer: We have made corrections by presenting the names of microbes in italics. See lines 423,425 and 427.